# A synthetic biochemistry platform for cell free production of monoterpenes from glucose

Tyler P. Korman[1], Paul H. Opgenorth[1] & James U. Bowie[1]

Cell-free systems designed to perform complex chemical conversions of biomass to biofuels or commodity chemicals are emerging as promising alternatives to the metabolic engineering of living cells. Here we design a system comprises 27 enzymes for the conversion of glucose into monoterpenes that generates both NAD(P)H and ATP in a modified glucose breakdown module and utilizes both cofactors for building terpenes. Different monoterpenes are produced in our system by changing the terpene synthase enzyme. The system is stable for the production of limonene, pinene and sabinene, and can operate continuously for at least 5 days from a single addition of glucose. We obtain conversion yields $>95\%$ and titres $>15\,g\,l^{-1}$. The titres are an order of magnitude over cellular toxicity limits and thus difficult to achieve using cell-based systems. Overall, these results highlight the potential of synthetic biochemistry approaches for producing bio-based chemicals.

[1] Department of Chemistry and Biochemistry, UCLA-DOE Institute, Molecular Biology Institute, University of California, Los Angeles, California 90095-1570, USA. Correspondence and requests for materials should be addressed to J.U.B. (email: bowie@mbi.ucla.edu).

Terpenes comprise one of the largest classes of industrially relevant bioactive natural products and have found wide use as flavours, fragrances, biofuels, commodity chemicals, vitamins and pharmaceuticals[1–5]. While many of the industrial and pharmaceutically relevant terpenes in use are plant-derived[6], low yields from natural sources limit their use and makes production subject to the unpredictability of agricultural boom and bust cycles[7,8]. Production in metabolically engineered microbes could therefore provide an attractive alternative for increased terpenoid production[9–12]. *Escherichia coli* and *Saccharomyces cerevisiae* have both been successfully engineered to produce monoterpenes such as limonene[13], pinene[14] and sabinene[15]; sesquiterpenes such as santalene[16,17], bisabolene[18] and amorphadiene[19]; and diterpenes such as sclareol[20], taxadiene[21] and carotinoids[22]. Nevertheless, the high titres and yields needed for economic viability of high-volume chemical uses have generally not been realized. Even in the case of monoterpenes limonene and pinene where established supply streams exist, their wider adoption as bio-safe solvents and biofuels is limited by amounts produced in nature[23]. Titres of terpenes are typically very low ($<0.5\,g\,l^{-1}$), and in even the best reported cases are only around $\leq 1.5\,g\,l^{-1}$ (refs 24,25). The notable exception is amorphadiene, which can be produced at $40\,g\,l^{-1}$ in *S. cerevisiae,* but the yields are too low (17% carbon yield) to be useful for low-cost chemicals such as biofuels[19].

Multiple factors limit the success of engineered microbes for terpene production (and biochemical production in general), including toxicity of products or intermediates, competing endogenous pathways, difficult pathway optimization (at both the transcriptional and translational level) and expensive product isolation from complex growth media[26]. In the case of monoterpenes limonene and pinene, a cellular toxicity limit of $\leq 0.5\% \,(5\,g\,l^{-1})$ is a major contributor to low titres and presents a key barrier for production in a microbial host[27,28].

Cell-free production provides an alternative approach to chemical transformations that can ease the technical challenges of engineering microorganisms and the limitations imposed by requiring cell viability[26]. Rather than engineer cells to produce and house the desired biochemical pathway, the cell-free approach simply mixes enzymes together in a reaction vessel (either from crude preparations or purified)[29–32]. While freedom from the constraints of the cell has many advantages, it also poses challenges because the complex regulatory systems and biochemical replenishing systems that exist in cells are eliminated. The design of enzyme systems that can be self-sustaining for long periods of time requires special considerations—a process we call synthetic biochemistry[32].

Here we use synthetic biochemistry to build a system for the production of monoterpenes from glucose. The system is stable and functions continuously for at least 5 days without any further additions of cofactors or enzymes. The high yields and titres for limonene ($12.5 \pm 0.3\,g\,l^{-1}$) and pinene ($14.9 \pm 0.6\,g\,l^{-1}$) are nearly an order of magnitude higher than that has been demonstrated in living systems. Our results suggest that commodity chemicals and pharmaceutical intermediates derived from acetyl-CoA and ATP are accessible through synthetic biochemistry systems.

## Results

**Overview of *in vitro* monoterpene biosynthesis from glucose.** The challenges facing synthetic biochemistry can be seen in the conversion of glucose to monoterpenes by a combination of the Embden–Meyerhof–Parnas (EMP) glycolytic and the mevalonate pathways. The mevalonate pathway requires 3 acetyl-CoA, 2 NADPH and 3 ATP to yield the terpene-building blocks isopentenyl pyrophosphate or dimethylallyl pyrophosphate. From 1.5 glucose, standard EMP glycolysis (hereafter referred to as 'glycolysis') will yield the 3 acetyl-CoA and the net 3 ATP needed, but will also produce 6 NADH, creating two problems: (1) an excess of reducing equivalents is produced; and (2) they are of the wrong type (NADH rather than NADPH) so that $NAD^+$ would not be recycled. We have previously developed a method to handle $NAD(P)H/NAD(P)^+$ balancing by creating enzymatic purge valve nodes[32,33] that use a combination of $NAD^+$- and $NADP^+$-utilizing dehydrogenases and NADH oxidase (NoxE) so

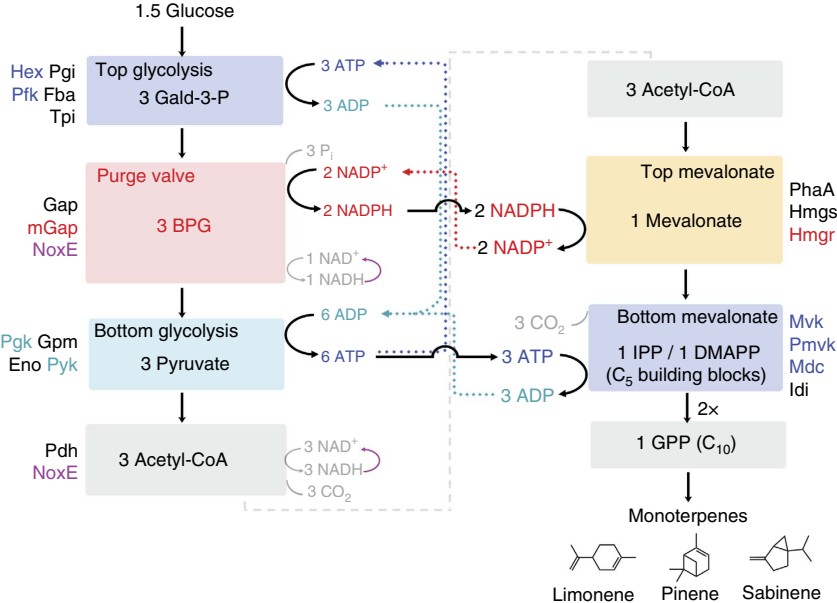

**Figure 1 | Schematic of the synthetic biochemistry system for conversion of glucose to monoterpenes.** Enzymes in each submodule (coloured boxes) are labelled as described in the text and Supplementary Table 1. ATP-consuming and -generating steps are shown in purple and blue, respectively. The NADPH-generating purge valve is shown in red while the NADPH-consuming top of the mevalonate pathway is shown in orange. Solid arrows show the flow of cofactors and dotted arrows highlight their recycle through respective steps in the pathway.

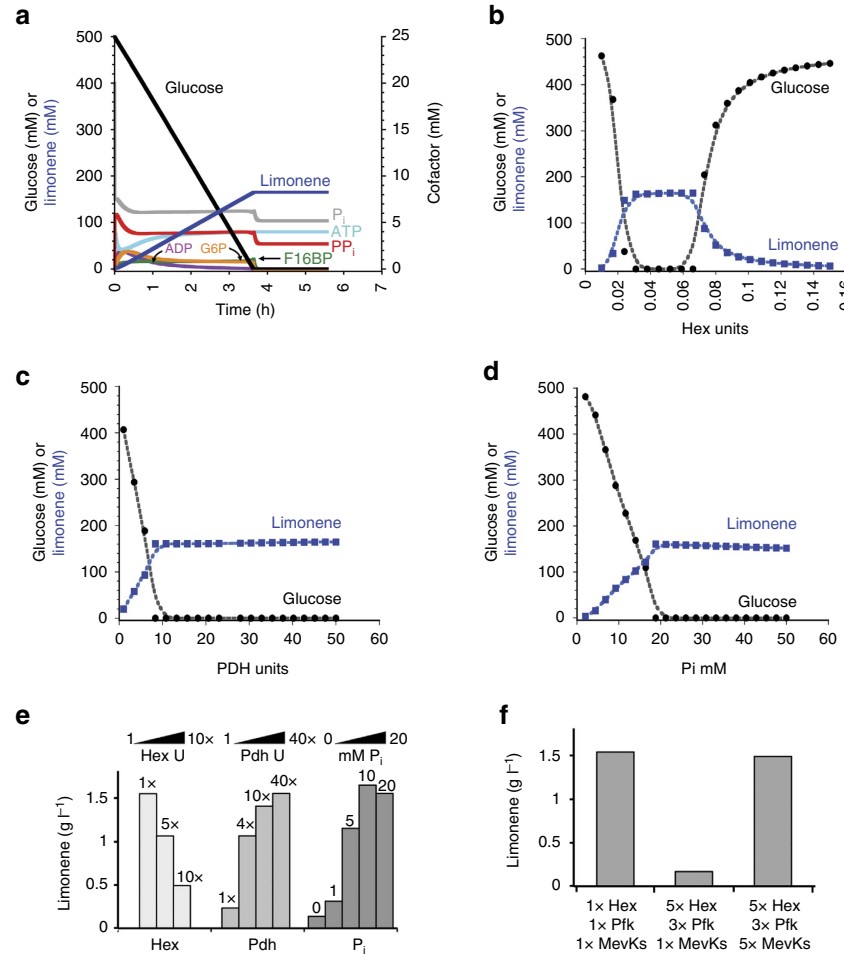

**Figure 2 | Modelling and optimization of enzyme levels for conversion of glucose to limonene.** (**a**) Levels for glucose (black line), limonene (blue line) and cofactors (ATP, ADP, G6P, F16BP, $P_i$ and $PP_i$) in an optimized CoPASI model starting at 500 mM glucose and a time of 20,000 s (5.6 h). Glucose and limonene levels are plotted on the left axis and cofactor levels are plotted on the right axis. (**b**) Parameter scan in CoPASI showing the dependence of limonene production on units of Hex varied between 0.05 and 0.15 units. (**c**) Parameter scan in CoPASI showing the dependence of limonene production on units of Pdh varied between 1 and 50 units. (**d**) Parameter scan in CoPASI showing the dependence of limonene production on starting $P_i$ varied between 1 and 50 mM. (**e**) Experimental data showing the dependence of limonene production on units of Hex, Pdh and $P_i$. (**f**) Experimental data showing the dependence of limonene production on the ratio of Hex and Pfk units to the kinases in the mevalonate pathway: Mvk; Pmvk; and Mdc (MevKs). The reactions in **c** and **d** were started with the addition of 250 mM glucose and analysed after 16 h.

that NADPH can be generated via the $NADP^+$-utilizing dehydrogenase, but excess reducing equivalents can be purged via NoxE to maintain carbon flux (see ref. 31 for description). The ATP pay-in phase of glycolysis presents an additional complication because if ATP is used too rapidly in the first hexokinase (Hex) step, there will be no ATP left to allow flux through the phosphofructokinase step[34]. Finally, high-energy phosphate for all metabolites (for example, the γ-phosphate of ATP) is ultimately (re)generated from inorganic phosphate ($P_i$), so that $P_i$ concentrations can affect flux through the system.

The goal of this study was to show that glucose breakdown through glycolysis can produce enough carbon building blocks and ATP to drive an important anabolic process (for example, the mevalonate pathway). So far, complex cell-free systems for building chemicals have either avoided the use of ATP completely[30] or stoichiometrically balance ATP use and consumption during catabolic reactions only[35]. While ATP-specific regeneration systems have been developed, these are generally simple systems consisting of only a few enzymes that utilize sacrificial substrates (for example, polyphosphate or acetyl-phosphate)[36,37]. It remains unclear whether excess ATP

(for example, the net 2 ATP produced from glycolysis) can be effectively used outside the cell to drive anabolic pathways in a fully recyclable manner[30,31]. In the synthetic biochemistry system we designed (Fig. 1 and Supplementary Fig. 1), ATP, NADPH and acetyl-CoA are generated using glycolysis reconstituted from purified components (Supplementary Table 1). NADPH production is regulated with a new molecular purge valve at the glyceraldehyde-3-phosphate dehydrogenase (Gap) step (rather than the pyruvate dehydrogenase (Pdh) step described previously[32]) by including a mutant Gap (mGap) that is specific for $NADP^+$ along with NoxE. Geranyl-pyrophosphate is ultimately produced through the canonical mevalonate pathway using the acetyl-CoA, ATP and NADPH output from the engineered EMP pathway. Various monoterpenes can then be made simply by adding different monoterpene synthases in the final step following geranyl-pyrophosphate production. Importantly, pyrophosphate released by the prenyltransferase and terpene synthase steps is then converted back to $P_i$ by the action of a pyrophosphatase, ensuring that all cofactors (ATP/ADP/NAD/NADP/CoA/$P_i$) are recycled continuously *in situ* and there is no build-up of any metabolite besides the

intended product. We also added catalase, glutathione and glutathione reductase to ensure the system maintains a reduced environment (Supplementary Table 1).

**Modelling with CoPASI to identify potential bottlenecks.** To ensure that our designed system could run effectively in theory, and to determine factors that may contribute to maximizing overall titres of limonene from glucose, we developed a computer model using CoPASI[38]. Although we have not measured kinetic parameters for the enzymes used experimentally, our goal was to explore fundamental features of the overall system design. We therefore used kinetic parameters defined for homologous enzymes where needed to obtain a reasonable model. Equations describing enzyme kinetics for glycolysis and the mevalonate pathway were taken from established models describing yeast metabolism[39] (Supplementary Note) with values for $K_m$, $V_{max}$ and $K_{eq}$ based on published values in the BRENDA database[40] and previous modelling experiments[39]. The Gap purge valve (including EcGap, mGap and NoxE) and pyrophosphatase were also included in the model. The final parameters and equations are provided in the Supplementary Note.

A virtual time course was performed starting with 500 mM glucose and system parameters (for example, $V_{max}$ or starting cofactor concentration) were adjusted to identify features important for the reaction to go to completion (final values shown in Supplementary Table 2). We focused on parameters that would allow the reaction to go to completion rather than improve reaction rate because we wanted to develop a system that was robust and sustainable, that is, could run for long periods of time without intervention (Fig. 2a). Modelling with CoPASI suggests that system sustainability is particularly sensitive to Hex (Fig. 2b) and Pdh enzyme activity (Fig. 2c), and also to initial $P_i$ concentration (Fig. 2d).

The importance of Hex and $P_i$ is perhaps unsurprising as Hex and $P_i$ are instrumental in ATP usage and recycling. It has been noted previously that too much Hex activity leads to a build-up of glucose-6-phosphate (G6P) and depletion of ATP, which is needed for the phosphofructokinase reaction[34]. ATP depletion can thereby block passage through the ATP pay in phase of glycolysis. Similarly, sequestration of $P_i$ in hexose phosphates (for example, G6P and fructose-1,6-bisphosphate) can block the Gap step, shutting down glycolysis (Supplementary Fig. 2a). The strong dependence of the reaction sustainability on Pdh activity is less clear. Since Pdh is followed by the first three ATP-independent steps of the mevalonate pathway (Fig. 1 and Supplementary Fig. 1), it is possible that too little Pdh causes a lag before ATP usage via Mvk, Pmvk and Mdc. As a result of this lag, Hex and Pfk might out-compete the kinases in the mevalonate pathway and again phosphate would become sequestered in hexose phosphates at the top of glycolysis. Overall, CoPASI modelling suggests that initial levels of Hex, $P_i$ and Pdh are the most critical contributors to successful conversion of glucose to limonene (Supplementary Fig. 2b).

**Implementation of _in vitro_ monoterpene production system.** Since the modelling results suggest that the system design could function in a sustainable fashion, we set out to implement the pathway. We set initial cofactor concentrations and the relative glycolysis enzyme levels as described in a prior work[41]. The rest of the enzymes levels were set so they would not be rate-limiting. We then varied Hex, Pdh or $P_i$ concentrations over a 10-fold range, keeping the rest of the system concentrations constant. Figure 2e shows the dependence of limonene titres on the amount of Hex, $P_i$ or Pdh input during a 16 h reaction. For $P_i$ or Pdh, the extent of reaction increases with increasing loading until

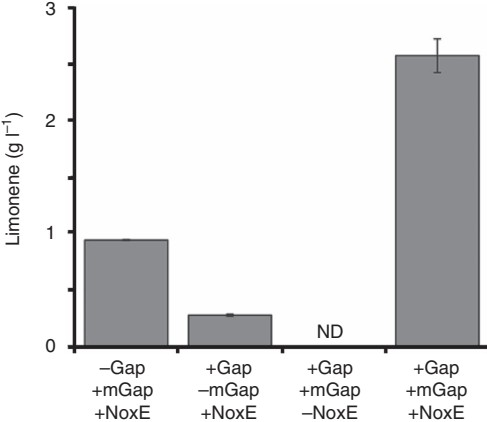

**Figure 3 | Dependence of limonene production on the new purge valve.** The graph shows the dependence of limonene production from glucose on the purge valve implemented at the Gap step. Various components were left out of each reaction (Gap, wild-type Gap from _E. coli_; mGap, NADP$^+$-specific mutant Gap from _G. stearothermphilus_; NoxE, NADH oxidase from _Lactococcus lactis_). The last experiment ($+$Gap, $+$mGap and $+$NoxE) contains all components. Leaving out any component leads to reduced limonene production from glucose. All reactions were started with the addition of 250 mM glucose and analysed after 16 h. Reactions were performed in triplicate ($n = 3$) and analysed as described in the Methods. Error bars represent s.d. ND, not determined.

a threshold is met, consistent with the COPASI modelling results. The limonene titre increased with increasing units of Hex activity to a certain point after which the limonene titre began to decline. If too much Hex was added, the rate of glucose phosphorylation outpaced Pfk and Mvk, Pmvk and Mdc activity. As shown in Fig. 2f, adding additional units of Pfk, Mvk, Pmvk and Mdc recovers limonene titre to starting conditions. The results suggest that if the kinase steps are not balanced, $P_i$ becomes sequestered in G6P, fructose-6-phosphate and fructose-1,6-bisphosphate (FBP; hexose phosphates), and Gap activity ceases. Clearly, care must be taken to balance Hex activity with all other kinases.

**Purge valve requirement for sustainable limonene production.** To test whether a molecular purge valve is necessary to drive the mevalonate pathway using glycolysis, we developed a new molecular purge valve at the Gap step (composed of an NAD$^+$-dependent Gap from _E. coli_, an NADP$^+$-dependent mGap from _Geobacillus stearothermophilus_ and NoxE) and evaluated the consequences of eliminating purge valve components. As shown in Fig. 3, leaving out any part of the new molecular purge valve significantly compromises the ability of the system to convert glucose into limonene. In fact, leaving out NoxE completely abolished limonene production from glucose as seen in prior characterization of other purge valves[32,33]. The dependence on NoxE is not surprising since both the molecular purge valve and Pdh rely on NoxE activity for unimpeded flux. With the full system, $2.57 \pm 0.15\,g\,l^{-1}$ (average of three independent measurements; $n = 3$) limonene could be produced in 24 h. Therefore, the new molecular purge valve at Gap can produce NADPH and maintain enough flux to provide ATP and acetyl-CoA for limonene production.

**Continuous production of limonene from glucose.** To test efficient and sustainable _in vitro_ production of limonene from glucose, we set up a semi-optimized system (Supplementary Table 3) and allowed the reaction to proceed for multiple days from a single addition of glucose. To monitor function and

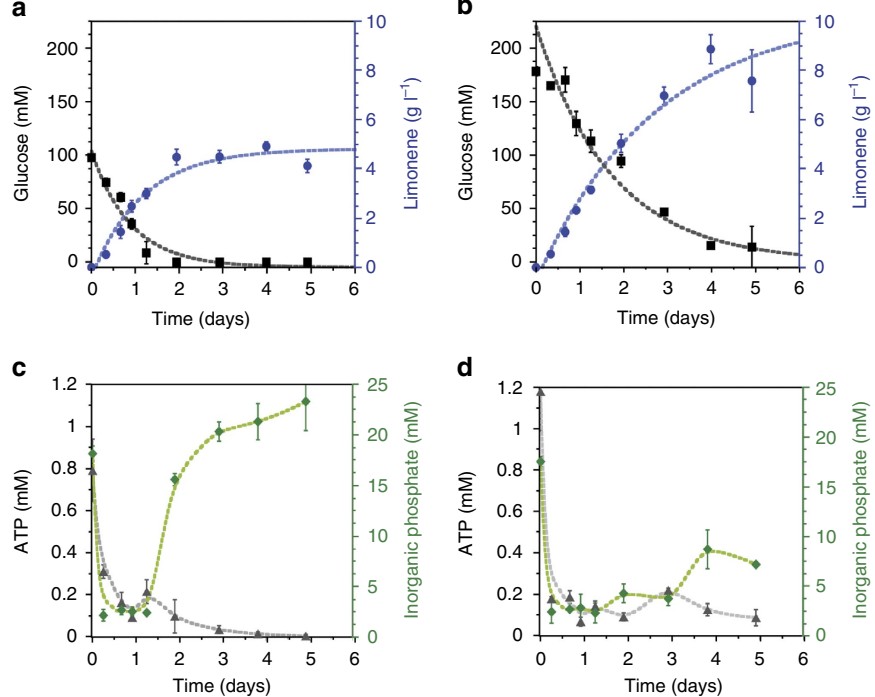

**Figure 4 | Time course of limonene production from glucose.** Time course of limonene production, glucose consumption, and ATP and $P_i$ levels starting from either 100 (**a,c**) or 200 mM (**b,d**) glucose over 5 days. The reaction starting from 200 mM glucose produces twice as much limonene over 5 days compared to starting with 100 mM glucose. Reactions were performed in triplicate ($n = 3$) and analysed as described in the Methods. Error bars represent s.d.

longevity, a single addition of 100 mM (Fig. 4a), or 200 mM glucose (Fig. 4b) was added at the start and the reaction was evaluated at various time-points for limonene production, remaining glucose and levels of ATP and $P_i$ (Fig. 4). Within 2 days, $4.94 \pm 0.18 \text{ g l}^{-1}$ ($n = 3$) limonene was produced from 100 mM glucose (Fig. 4a), which corresponds to a 100% theoretical yield at a maximum productivity of $\sim 0.1 \text{ g l}^{-1} \text{h}^{-1}$. Monitoring glucose (total hexose and hexose phosphates), ATP and $P_i$ show that all glucose input is consumed and ATP and $P_i$ cycle accordingly (Fig. 4c). When twice as much glucose was input (200 mM), glucose consumption and limonene production rates were nearly constant over 5 days reaching $8.87 \pm 0.59 \text{ g l}^{-1}$ ($n = 3$) limonene and correspond to a theoretical yield of 97.6% (Fig. 4b). When ATP and $P_i$ were monitored concurrently (Fig. 4d), both ATP and $P_i$ levels decrease to a low basal level ($\sim 200 \,\mu\text{M}$ and 2 mM for ATP and $P_i$, respectively) and then fluctuate over the course of 4 days as would be expected (that is, when ATP is higher, $P_i$ is lower and vice versa). When 200 mM glucose was input, the limonene titre did not plateau until day 4 when all of the available glucose had been consumed, suggesting the system could progress further. Finally, to determine the full extent of biosynthetic capability for the *in vitro* system, 500 mM glucose was input and limonene production monitored over 7 days. At day 7, limonene reached a final titre of $12.5 \pm 0.3 \text{ g l}^{-1}$ ($n = 3$; Fig. 5) with a total yield of $88.3\% \pm 4.6\%$ ($n = 3$) of theoretical suggesting little or no loss of carbon due to side reactions. Our *in vitro* production of $12.5 \text{ g l}^{-1}$ limonene directly from glucose is nearly an order of magnitude higher than has so far been demonstrated for limonene production in living systems[42] and $>2$ times the toxicity limit of between 0.02 and 0.5% for *E. coli* or *S. cerevisiae*[27,43].

**Production of other monoterpenes from glucose.** To test whether our synthetic biochemistry system could be used as a broad platform for the production of other monoterpenes, we

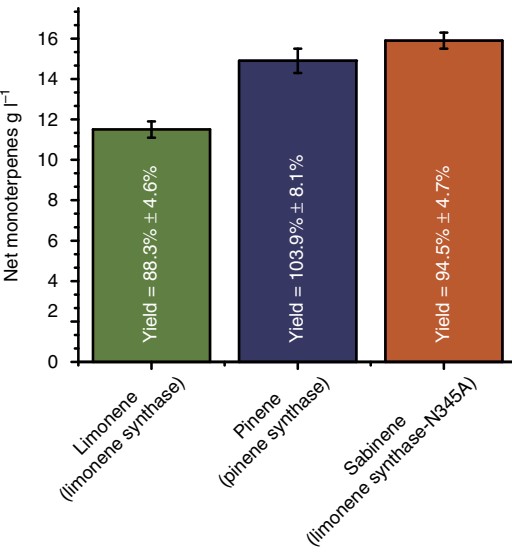

**Figure 5 | High-titre production of multiple monoterpenes from glucose.** The chart shows production of monoterpenes limonene, pinene and sabinene from 500 mM glucose over 7 days. The final titre of monoterpenes and glucose remaining were determined by GC and enzyme assay as described in the Methods. For the limonene synthase reaction (green bar), the product was 100% limonene. For the pinene synthase reaction (purple bar), the product distribution is $80.8 \pm 0.33\%$ α-pinene and $19.2 \pm 0.33\%$ β-pinene. For the sabinene synthase reaction catalysed by the N345A Limonene synthase mutant (orange bar), the product distribution is $11.5 \pm 0.03\%$ α-pinene, $11.2 \pm 0.01\%$ β-pinene, $37.9 \pm 0.02\%$ ( − )-sabinene, $3.5 \pm 0.01\%$ ( + )-sabinene, $27.1 \pm 0.04\%$ limonene and $5.7 \pm 0.01\%$ myrcene. Reactions were performed in triplicate ($n = 3$) and analysed as described in the Methods. Error bars represent s.d.

attempted to produce pinene and sabinene from glucose. To make pinene or sabinene, the appropriate monoterpene synthase was simply swapped with the limonene synthase used above. For pinene synthesis, α-pinene synthase from *Picea sitchensis* was used, which has been shown to produce a mixture of 77% α-pinene and 22% β-pinene[44]. For sabinene production the limonene synthase mutant N345A was used, which produces sabinene as its major product along with smaller amounts of other monoterpenes[45]. Starting with 500 mM glucose, monoterpene production was allowed to proceed for 7 days and then quantified by GC (Supplementary Fig. 3). Pinene titre (sum of α-pinene and β-pinene nearly identical to distribution in ref. 44) reached $14.9 \pm 0.6\,\mathrm{g\,l^{-1}}$ ($n = 3$) while sabinene titre (total monoterpenes, similar distribution to ref. 45) was $15.9 \pm 0.4\,\mathrm{g\,l^{-1}}$ ($n = 3$) with yields of $103.9\% \pm 8.1\%$ ($n = 3$) and $94.5\% \pm 4.7\%$ ($n = 3$) of theoretical, respectively (Fig. 5). Like limonene production, these values are at least an order of magnitude higher than microbial production for pinene[14] and sabinene[15] and multiple times higher than the toxicity limit for these compounds[15,43].

## Discussion

Our results show that industrially relevant complex biomolecules such as monoterpenes can be produced *in vitro* directly from glucose using a synthetic biochemistry platform. Specifically, we find that the ATP and the carbon-building block acetyl-CoA generated by glycolysis can be used to drive the biosynthesis of complex compounds directly from glucose, a low-cost, renewable feedstock. This represents a significant expansion of synthetic biochemistry platform as both acetyl-CoA and ATP are needed for the biosynthesis of many desirable products besides terpenes. We were able to produce monoterpenes at high titres ($12.5 \pm 0.3$, $14.9 \pm 0.6$ and $15.9 \pm 0.4\,\mathrm{g\,l^{-1}}$ for limonene, pinene and sabinene, respectively) and yields ($88.3 \pm 4.6\%$, $103.9 \pm 8.1\%$ and $94.5 \pm 4.7\%$ of theoretical, respectively), and production was continuous over many days without any further additions of enzyme or cofactors. While the flux reported here is still about an order of magnitude below rates needed for industrial production ($0.1$ versus $1\text{–}2\,\mathrm{g\,l^{-1}\,h^{-1}}$), this work significantly improves on a previous, less-complicated system[41] from phosphoenolpyruvate (PEP) to isoprene in both rate ($0.014\,\mathrm{g\,l^{-1}\,h^{-1}}$) and overall extent of reaction (for example, titre and scale). Rates for *in vitro* production of less-complex molecules such as ethanol[34,46], 2,3-butanediol[47], mevalonate[31] and polhydroxybutyrate[33] from glucose in either a lysate or purified system range from 0.7 to as high as $47.9\,\mathrm{g\,l^{-1}\,h^{-1}}$, suggesting that the rate for *in vitro* terpene production could be raised if either more terpene synthase were used or the rate was significantly improved through engineering. Therefore, with further optimization of reaction conditions as well as *in vitro* evolution of enzyme stability and activity (for example, terpene synthase), it should be possible to develop a faster system that functions for much longer.

Without the limitations of toxicity and substrate competition, it should also be possible to expand the synthetic biochemistry platform to make sesquiterpenes and diterpenes simply by changing two proteins. Because the system works with an excess of NADPH, it is also possible to envision the incorporation of modifying enzymes such as cytochrome P450s to further expand the number and diversity of products generated.

Our results suggest that synthetic biochemistry has the potential to reach industrially relevant production parameters for a wide range of complex molecules that are difficult to produce *in vivo*. The key will be to keep enzyme costs low by finding stable enzymes so they can be used for long periods of time, methods for recycling the enzymes and inexpensive purification methods. Implementing a more stable system with engineered enzymes that last for weeks or months will significantly lower the cost of the enzymes not only in terms of dollars but also in initial input of glucose needed to produce enzymes (which was not taken into consideration for our yield calculations). The longer the proteins last (that is, higher total turnover number), the larger the reduction in up-front costs, including glucose. Our monoterpene platform demonstrates that *in vitro* systems can reach yields and titres required for industrial production of specialty bio-based chemicals but significant improvements are necessary in areas such as stability and total turnover number before synthetic biochemistry can begin to challenge synthetic biology for production of commodity chemicals from biomass.

## Methods

**Chemicals and reagents.** All buffers, reagents and cofactors (for example, bis-tris propane, glucose, limonene, pinene, sabinene, FBP, ATP, NAD(P)$^+$, coenzyme A and so on) were purchased from Sigma-Aldrich. Yeast Hex (H6380) was from Sigma. Hexanes, isopropyl myristate and nonane were from ARCOS and were the highest grade available. Liquid and solid media for growth of *E. coli* was from Fisher Scientific (BD Difco).

**Cloning and purification of enzymes.** All enzymes were cloned into pET28a(+) (Novagen) with an N-terminal 6xHis-tag unless otherwise noted (Supplementary Table 1). NCBI gene accession codes used in this work are listed in Supplementary Table 1. Primers used for cloning are listed in Supplementary Table 4. *E. coli* strain BL21Gold(DE3) (Agilent) was used for both cloning and overexpression of His-tagged proteins. Genes were amplified from plasmid or genomic DNA using HotStart Taq Mastermix (Denville) and then cloned into PCR-amplified vectors using a modified Gibson method as described previously[41,48]. Mutagenesis of limonene synthase to introduce the N345A mutation[45] and GsGap to construct the D32A/L33R/T34K triple mutant was done using the Quickchange Site-Directed Mutagenesis Kit (Stratagene).

For overexpression and purification of all constructs (except Pdh, see below), 1 l of Luria broth (LB) media containing antibiotic was induced with 0.4 mM isopropyl-β-D-thiogalactoside once the optical density 600 nm reached 0.6. After 16 h incubation, soluble 6xHis-tagged proteins were purified from clarified cell lysates by Ni-NTA affinity chromatography, adjusted to 10% glycerol, flash-frozen with liquid $N_2$ and stored at $-80\,°C$ until use. For terpene synthases, enzymes were concentrated to $\sim 20\,\mathrm{mg\,ml^{-1}}$ with a 30 kDa molecular weight cut-off concentrator (Amicon) before freezing.

**Assembly and purification of Pdh complex.** All three subunits of the Pdh complex (AceE, AceF and Lpd) were cloned as N-terminal 6xHis-tag constructs as above. The AceE and Lpd subunits of Pdh were expressed and purified individually by Ni-NTA. For AceF overexpression, the media was supplemented with 0.2 mM (±) lipoic acid before induction with isopropyl-β-D-thiogalactoside. Following lysis, AceF was precipitated by two additions of ammonium sulfate to 25 and 55%. The pellet from the 55% ammonium sulfate addition was dialysed into 25 mM Tris-Cl (pH 8.5), and purified by ion-exchange chromatography using a 25 ml HiTrap Q column. Fractions containing AceF were pooled, and AceF was isolated by ultracentrifugation ($105,000\,g$ for 3 h). The pellet containing AceF was resuspended in 25 mM Tris-Cl (pH 8) containing 10% glycerol and then mixed with Ni-NTA-purified AceE and Lpd. The mixture was allowed to incubate at $4\,°C$ for 30 min to form the complex and then isolated by ultracentrifugation ($105,000\,g$ for 3 h). The resulting yellow pellet was resuspended in 7 ml 25 mM Tris-Cl (pH 8) and 0.1 M NaCl containing 50% glycerol, and stored at $-20\,°C$.

**CoPASI modelling.** CoPASI v4.16 was downloaded from http://copasi.org/ (ref. 38). For simulation and analysis of the combined glycolytic and mevalonate pathways designed in this study, $K_m$ and $k_{cat}$ values were obtained from the BRENDA database[40] (Supplementary Table 2) and rate equations (Supplementary Note and Supplementary equations (1)–(23)) were derived based on descriptions of yeast metabolism[39]. A time course task was run with a timescale of 20,000 s and parameters adjusted until a steady rate of conversion of glucose into limonene was obtained. Next, a parameter scan task was performed for the $V_{max}$ of various enzymes or starting concentration of various cofactors (for example, $P_i$ or ATP) over either a 10- or 50-fold range in 20 steps to identify bottlenecks and optimum starting conditions. Bottleneck enzymes (for example, Pdh) were identified by starting with a 250- to 500-fold excess over Hex and systematically decreasing each enzyme to a 5- to 10-fold excess followed by monitoring limonene production over 20,000 s. Pdh was the only enzyme that failed to run to completion during the given time frame.

**Enzyme assays.** Enzymes used in this work were assayed as described previously[32,33,41]. In general, enzymes that produce or consume NAD(P)H were monitored at 340 nm. Enzymes that do not produce or consume NAD(P)H directly were coupled to an enzyme that does (for example, FBP aldolase coupled to glyceraldehyde-3-phosphate dehydrogenase). Enzymes that consume or produce ATP were monitored at 340 nm using a coupled assay to NADH consumption through the activity of pyruvate kinase and lactate dehydrogenase. All readings were performed in triplicate ($n = 3$) using a SpectraMax M5 Plate Reader.

**Initial limonene production from glucose.** Initial reactions for the continuous *in vitro* bioconversion of glucose to limonene were composed of 100 mM Tris-Cl (pH 7.5), 6 mM MgCl, 10 mM KCl, 2 mM ADP, 2 mM ATP, 1 mM FBP, 1.5 mM CoA, 0.5 mM NAD$^+$, 1.5 mM NADP$^+$, 100 mM glucose, 0.5 mM thiamine pyrophosphate, 0.25 mM 2,3 bisphosphoglycerate, 5 mM reduced glutathione and 10 mM $P_i$ in a final volume of 200 µl. Glutathione reductase and catalase were simply added as a precaution to prevent oxidation over the course of 7 days. We did not optimize or investigate the need for these components. As the overall system design with the purge valve produces an excess of NADPH, there is sufficient NADPH in the system to supply glutathione reductase. The enzyme concentrations are given in Supplementary Table 3. The reactions were initiated with the addition of glucose followed by carefully overlaying the reaction sample with 600 µl isopropyl myristate. The reactions were allowed to proceed for at least 24 h and gentle shaking with limonene production monitored by gas chromatography equipped with a flame ionization detector (GC-FID).

Initial levels of glycolysis enzyme loadings were set based on data from Scopes and Welch[34] for *in vitro* ethanol production from glucose. Levels for Pdh and mevalonate pathway enzymes were set based on measured (PhaA, Hmgs, Hmgr, Mvk, Pmvk and Mdc) or reported (Idi and Fpps) units per milligram values such that none of the mevalonate pathway enzymes (with the exception of limonene synthase) were limiting. Minimal optimization was performed by varying key enzyme (Hex, Pdh, NoxE, Gap and mGap) and cofactor (NAD$^+$, ATP and phosphate) levels until a maximum titre could be established in 24 h that was dependent on the terpene synthase activity. Final enzyme loadings are listed in Supplementary Table 3.

**Continuous terpene production from glucose.** Reactions were set up in triplicate at a volume of 200 µl in 2 ml glass vials and incubated uncapped with an organic overlay at 25 °C and gentle shaking. Each reaction contained 100 mM bis-tris propane (pH 7.5), 20 mM potassium phosphate (pH 7.5), 1 mM FBP, 4 mM ATP, 1.5 mM NADP$^+$, 0.2 mM NAD$^+$, 1.5 mM CoA, 6 mM MgCl2, 10 mM KCl, 0.5 mM thiamine pyrophosphate, 0.25 mM 2,3 bisphosphoglycerate and 5 mM glutathione. For limonene and pinene production, *Gs*Fpps mutant S82F was used in combination with *Mentha spicata* limonene synthase (produces 100% 4S-limonene) or *P. sitchensis* α-pinene synthase (produces 5:1 ratio of α and β pinene), respectively. For sabinene production, *M. spicata* limonene synthase mutant N345A was used (which produces sabinene as its major product along with smaller amounts of five other monoterpenes). The total catalyst load is detailed in Supplementary Table 3. Reactions were initiated with the addition of 125, 250 or 500 mM (final) glucose and then immediately overlaid with 600 µl isopropyl myristate. Reactions were allowed to proceed for the time indicated, vortexed for 5 s and centrifuged for 2 min at 16,060 × g to separate the aqueous and organic layers. For analysis and quantification of terpene production, 75 µl of the organic layer was mixed with 175 µl hexanes and analysed by GC-FID. For analysis of residual glucose, ATP and phosphate, the bottom aqueous layer was separated to a new tube, brought back to 200 µl with buffer and heat-treated for 2 min at 95 °C to denature proteins. Samples were then centrifuged (16,060 × g for 2 min) and stored at −20 °C until analysis.

The yield was calculated by dividing the moles of monoterpene produced by the theoretical moles of monoterpene that could be produced from the moles of glucose that entered the system (that is, glucose consumed, not total glucose input). The amount of glucose that entered the system was calculated by the amount of glucose measured at $t = 0$ minus the amount of glucose remaining at $t = 7$ days. All reactions were performed in triplicate ($n = 3$).

**Analytical methods.** Terpenes isolated in the organic layer were analysed by GC-FID (HP5890II) equipped with a HP-INNoWax column (0.32 mm × 30 m, Agilent). A volume of 1 µl was injected and terpenes were identified and quantified by comparison to an authentic standard curve (limonene, α-pinene or sabinene). Representative traces are shown in Supplementary Fig. 3. The carrier gas was helium with a flow rate of 5 ml min$^{-1}$. The oven temperature was kept at 70 °C for 3 min, raised to 260 at 5 °C min$^{-1}$ and then held at 260 °C for 3 min. The inlet and detector temperatures were kept at 250 and 300 °C, respectively.

For analysis of glucose, a coupled enzymatic assay was performed based on reduction of NAD$^+$ to NADH by Gap. Briefly, heat-treated samples were diluted and 2 µl was added to 200 µl assay mix containing 100 mM Tris-Cl (pH 7.5), 5 mM MgCl2, 10 mM KCl, 1 mM ATP, 2 mM NAD$^+$, 7 mM $P_i$ and fresh Hex, Pgi, Pfk, Ald, Tpi, Gap and Pgk. The change in absorbance was monitored for 10 min at 340 nm. The amount of glucose present was calculated based off a standard curve from known quantities of glucose.

For analysis of ATP, the ATP Bioluminescent Assay Kit (Sigma-Aldrich) was used. Before performing the quantification, the ATP Assay Mix was diluted 625-fold with ATP-Assay Mix Dilution buffer. To quantify ATP, 100 µl of a 1:100 diluted sample was added to 100 µl ATP Assay Mix and the luminescence was measured immediately. A standard curve for ATP was run concurrently and used to calculate the amount of ATP present.

For analysis of $P_i$, the Malachite Green Phosphate Detection Kit (R&D Systems) was used. A volume of 1 µl of a 1:100 dilution of sample was added to 49 µl H$_2$O and then processed according to the manufacturer's directions. Phosphate was calculated by comparison to a standard curve run concurrently with known amounts of $P_i$.

**Data availability.** NCBI gene accession codes used in this work are listed in Supplementary Table 1. All the other data supporting the findings of this study are available within the paper and its Supplementary Information files and from the corresponding author on reasonable request.

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

## Acknowledgements

We thank members of the lab for helpful comments. This work was supported by DOE grant DE-FC02-02ER63421 and ARPA-E DE-AR0000556 to J.U.B.

## Author contibutions

All the authors contributed to the conception and planning of the project. T.P.K. and P.H.O. performed the experiments. All the authors analysed the results. T.P.K. and J.U.B. wrote the manuscript.

## Additional information

**Competing Financial Interests:** The authors have formed a company, Invizyne Technologies, to expand and develop cell-free biomolecule production technologies.

**Reprints and permission** information is available online at http://npg.nature.com/ reprintsandpermissions/

