## [Peer Review File · Nature Communications]

Reviewer #1 (Remarks to the Author)

The manuscript „A synthetic biochemistry platform for cell free production of monoterpenes from glucose” by Korman et al is an impressive piece of cascade reaction engineering. A total of 27 purified enzymes are used to convert glucose via glycolysis to acetyl CoA and then via the mevalonate pathway to monoterpenes. The enzymes are purified separately and then used to assemble a fully defined and well-controlled reaction systems without side reactions. Some efforts are undertaken to optimize the system by setting up a model in copasi and testing for the influence of varying enzyme levels one by one relative to a reference state obtained from previous work. The suggestions from this modelling step are followed up in real experiments and found by and large confirmed. Importantly, varying the levels of hexokinase and pyruvate dehydrogenase as well as the phosphate concentration, the authors could establish a local optimum for monoterpene production.

The authors also implement one of their earlier achievements, the NADH purge valve. They can show that this purge valve is necessary to overcome imbalances in the NADPH regeneration. The authors run the optimized system for limonene production then for increasingly longer periods of time (up to 5d) and increasingly higher concentrations of substrate (up to 500 mM of glucose) and achieve full conversion. Finally, they demonstrate that, depending on the monoterpene synthase they use, they can also make two more monoterpene products, pinene and sabinene.

The authors document a number of very notable achievements: They report final product titers that are far higher than what has been documented with in vivo cultures (with the notable exception of amorphadiene), yields typically far beyond 80% (however, see below), operation times of up to 5 days, and flexibility in view of the product. They achieve cofactor regeneration for ATP, NADPH, and acetyl-CoA in their system. All of these are exceptional results for the implementation of enzyme networks, and in this case far better than nearly all the results that have been achieved in vivo. But also for in vitro systems (of comparable complexity or not), these are very notable achievements. Although larger networks have been reported in the literature, they were nowhere near this level of refinement, reflected in the yields and final product concentrations.

The impact of these results is somewhat mitigated that the authors published a recent report along very similar lines (different enzyme system including PPP enzymes for the production of polyhydroxybutyrate, similar yields, higher productivity, shorter operating times). However, I argue that the improvement above the general state of the art is so high and that the product group targeted in the present publication (isoprenoids, one of the central natural product classes) is so attractive, that I think that publication in NCOMMS is perfectly justified.

In a revised version, the authors should address the following issues:

The authors mention very impressive yields, but I could not find any indication of how these yields were calculated. Given that they use – for example – 200 mM of glucose (36 g/L) to produce for example 9 g of limonene, they should be explicit about how this yield was calculated.

The authors should at some point mention the entire catalyst load (eg in terms of g protein per liter) of one of their optimized systems. If I understand correctly, the catalyst load is somewhere on the order of 1 or 2g/L, which is impressively low. Most of this is pinene synthase and limonene synthase (I believe, the abbreviations used in supplemental table 3 are not well defined), which means that the supporting network is even far less concentrated.

I could not follow the authors in how they did their modelling steps (lines 110 to 115). Are they suggesting that they used the parameters for yeast enzymes even if they used in fact a different enzyme? If so, this should be justified. Any ambiguity could also be eliminated by completing supplemental table 2 with the used parameter values. The authors should also summarize in a separate supplemental table what parameter ranges they checked for the other enzymes that are

not mentioned in Fig. 2 in which step size and what their criterion was to decide that the influence was not worth pursuing further.

The authors might also want to specify further what they mean by “industrially relevant monoterpenes”. For example pinene is, to the best of my knowledge, a cheap byproduct of the coniferous wood industry, which is sold for a few dollars per liter (if that much). I doubt that this is more than an illustration product?

However, the few points mentioned above should not detract from the fact that I consider this contribution a major advance in the field of “synthetic biochemistry”, as the authors call it.

Reviewer #2 (Remarks to the Author)

The manuscript by Korman, Oppenorth and Bowie is certainly an eye-opener for me and I suspect it could be for many others. The Bowie lab has been pioneering their Synthetic Biochemistry system for a few years now and it does in fact seem to rival the Synthetic Biology platforms that are the current rage. The Synthetic Biochemistry platform looks elegant from the perspective of simply adding enzymes and the means for balancing co-factor needs, and viola, the system pumps out desired end-product with high fidelity and high efficiency.

Recent articles from the Bowie group have already described and demonstrated the importance of the NAD/NADP purge value node, so that is not new in the current work. But capturing this in combination with balancing ATP/ADP metabolism does demonstrate how multiple co-factors can be juggled simultaneously in a single system to give exceptional high conversions of feedstock to final product.

However, I do have a few quibbles. The authors probably appreciate that terpene synthases typically have very low turnover rates. Moreover, the reactions catalyzed by these enzymes are more akin to little explosion – rip off a diphosphate from the allelic substrate, GPP, with a highly reactive carbocation running around within the active site. There is some evidence in the literature that these enzymes actually catalyze maybe 2 to 4 cycles of catalysis in vitro before pooping out. Looking at the incubation conditions, there isn't anything to suggest the authors have done anything special to “protect” the enzymes. So, what I would like the authors to comment about is how much terpene synthase enzymes they are adding and what percentage of the enzyme is actually staying active during this time frame of days. Perhaps a calculation for the number of turnovers needed for the final yields in relationship units of activity added would be informative. I actually expect that they have to add a stoichiometric excess of the terpene synthases in order to observe the levels of monoterpenes produced. Some additional comments/calculations would be really helpful, especially because those who have used these enzymes in microfluidic devices tend to see the enzymes die within minutes to hours. What's the trick here?

When I examine the experimental conditions a bit more closely, I'm also struck that the experiments are being performed on a 200 μ l scale, yet the data is being expressed on a liter basis. If the authors have not confirmed that these assays can actually be scaled to these levels, then I strongly recommend caution in representing the data in these terms. (Also need more clarification about the overlay – what is the solvent and is there any special means for overlaying the solvent? Probably should provide more background on the various mutants used and give a reference for each.)

My most critical comment concerns the efficiency calculations. The authors are attempting to distinguish the Synthetic Biochemistry platform relative to the Synthetic Biology ones. And the calculation is for input glucose conversion to end-product monoterpene. In the Synthetic Biochemistry system, this is a relatively trivial calculation and improving efficiency is a matter manipulating the input enzymes and co-factors. But what is not being considered in these calculations is the “cost” of the enzymes. These were all produced in bacteria and there isn't any

consideration for the carbon feedstock used to grow those bugs. If one were to take those “costs” into consideration, I suspect the efficiency yield calculation would decline by an order of magnitude. The Synthetic Biology platform is appealing because it does not require all the effort of isolating enzymes, qualifying the isolated enzymes, and preserving them for use of days, months and years. Stocking only the bugs in a freezer serves to preserve the Synthetic Biology system for long-term future use. There are many such reasons why the Synthetic Biology platforms has appeal and offer major advantages over the Synthetic Biochemistry platform.

Overall, the experimental work presented has been done well and the story line is succinct and easy to digest. However, I do not buy that the Synthetic Biochemistry platform lends itself to large-scale, commodity chemical production. There is a niche for this platform and it may be small scale, perhaps focused on screening mutant enzymes for novel chemistry. Even for a short communication, this reviewer feels it is incumbent upon the authors to fairly represent their work relative to the competing technology.

Reviewer #3 (Remarks to the Author)

Summary

The manuscript by Korman et al. seeks to construct a purified enzymatic system for the synthesis of monoterpenes from glucose. This work builds off previous innovations from the authors in the discovery of metabolic purge valves, but extends those efforts in a new direction to a novel synthetic biochemical system. To demonstrate the generality of their approach, the authors present several vignettes to make limonene, pinene, and sabinene at unprecedented yields, to my knowledge. The combination of models and experiments further increased impact. The manuscript is well written, the work is original, and the concept is clearly explained. Indeed, this work synthesizes known and new information into a comprehensive story that has an exciting take home message that is certain to open many new avenues of research. For example, the work provides a new foundation to start thinking more closely about redox and the potential redox balance has to further drive cell-free production for prototyping and biomanufacturing. I support publication, pending some revisions.

Major claims of the paper

- The authors produce unprecedented amounts of monoterpenes in their in vitro purified enzyme system
 - Pinene titer (sum of α and β pinene) reached 14.9 ± 0.6 g/L
 - sabinene titer (total monoterpenes) was 15.9 ± 0.4 g/L
 - limonene reached a final titer of 12.5 ± 0.3 g/L
- Surprisingly, their system is stable and functional for multiple days (can exceed 5 days).
- A key feature of their system is a new “valve” enzyme, a mutant Gap (mGap) that is specific for NADP⁺, which regulates NADPH production.
- In addition, they developed a computational model (CoPASI) based on current kinetic data for each enzyme in their pathways. They use this to inform their optimization.

Concerns

- While their work is convincing, a full time course should be reported for figure 5. They mention a key limitation is their rates: “While the flux is still about an order of magnitude below rates needed for industrial production (0.1 g/L/hr vs 1-2 g/L/hr), this work significantly improves upon a previous, less complicated system from PEP to isoprene in both rate (0.014 g/L/hr) and overall extent of reaction (e.g. titer and scale).” However, they don’t provide us with data to visualize this ourselves. A full time course of the 7-day reactions for each monoterpene would significantly enhance the reader’s ability to observe this key limitation and raise the impact of the work.
 - Along these lines, I agree that the flux/rates through their pathway are low. The authors should more comprehensively compare these rates to other in vitro systems, both synthetic enzymatic pathways as well as lysates? What are strategies to improve the rates? Also, what do rates look like for different classes of molecules in various purified and lysate systems?

- Additionally, please represent figure 5 with appropriate distribution of alpha versus beta pinene. Also describe the relative amounts sabinene compounds (instead of summing total monoterpenes). The current plot implies that pure compounds are produced. Please comment how this system affects the ratio of alpha and beta pinene compared to reference 39.
- In Figure 3, the authors show a very simple +/- system of the valve enzymes. I am curious to see whether there are optimal amounts of these enzymes together. Are there concentrations of NoxE for example that still yield limonene but are low? Are there concentrations that are too high? How robust is this valve? While the authors use their computer model to identify bottlenecks and only choose to focus on the most sensitive parameters to emphasize robustness, not optimums, it would nice to see a sensitivity analysis or optimization of their new purge valve (as this is a key claim of their work)
- Additionally, how was the level of glutathione reductase chosen? This reaction consumes NADPH but the authors do not comment on its role in the system and how it interacts with the purge valve system.
- The system is described as “robust” in line 120 and 171. I suggest a brief sensitivity analysis to enzyme concentration and cofactor concentration [possibly also reaction temperature, volume, etc] to support this statement.
- Line 94...The authors state that “cell free systems have avoided the use of ATP for building chemicals” and cite a 2012 review by Guterl et al. However, high ATP turnover has been observed in a number of cell free systems, both purified and crude extracts, for synthesis of metabolites and proteins (e.g., Ye, X., Honda, K., Sakai, T., Okano, K., et al., Synthetic metabolic engineering-a novel, simple technology for designing a chimeric metabolic pathway. *Microb. Cell Factories* 2012, 11, 120 and others). Can the above statement be expanded to convey the idea that ATP regeneration has been used in cell-free systems before?
- While beyond the scope of this work, I am curious to know what are the limits of the current system? Why does the reaction stop? Is fed-batch feeding of glucose possible?

Minor concerns

- Please add a brief comment at line 98 or 159 to clarify that this purge valve is at the Gap step in contrast to previous work that made a purge valve at the PDH step (reference 31). This will help readers not intimately familiar with prior literature in the field.

Reviewer #1 (Remarks to the Author):

In a revised version, the authors should address the following issues:

1. The authors mention very impressive yields, but I could not find any indication of how these yields were calculated. Given that they use – for example – 200 mM of glucose (36 g/L) to produce for example 9 g of limonene, they should be explicit about how this yield was calculated.

RESPONSE: The calculation for yield was initially located in the figure legend for Figure 5. To be more explicit and easier to find, the description of yield calculation was moved to the materials and methods section under the heading “Continuous Terpene Production from Glucose.”

Added text:

“The yield was calculated by dividing the moles of monoterpene produced by the theoretical moles of monoterpene that could be produced from the moles of glucose that entered the system (i.e., glucose consumed, not total glucose input). The amount of glucose that entered the system was calculated by the amount of glucose measured at t=0 minus the amount of glucose remaining at t=7 days. All reactions were performed in triplicate (n=3).”

2. The authors should at some point mention the entire catalyst load (eg in terms of g protein per liter) of one of their optimized systems. If I understand correctly, the catalyst load is somewhere on the order of 1 or 2g/L, which is impressively low. Most of this is pinene synthase and limonene synthase (I believe, the abbreviations used in supplemental table 3 are not well defined), which means that the supporting network is even far less concentrated.

RESPONSE: Thank you for this suggestion. We have added stock enzyme concentrations, amount of enzyme added (in addition to total sum), and total turnover number (see Reviewer #2 comments below) to Supplemental Table 3. A note was also added at the bottom of the table referring the reader to Supplemental Table 1 for a more detailed description of enzyme names/abbreviations. As you note, the terpene synthases are indeed slow. The total enzyme loading are 6.6 g/L and 3.8 g/L without terpene synthase.

3. I could not follow the authors in how they did their modelling steps (lines 110 to 115). Are they suggesting that they used the parameters for yeast enzymes even if they used in fact a different enzyme? If so, this should be justified. Any ambiguity could also be eliminated by completing supplemental table 2 with the used parameter values.

RESPONSE: Yes, we should have been clearer. It would be impossible to experimentally measure all the required parameters. But our goal was not to precisely model our specific system, but to explore the overall system behavior. We therefore obtained reasonable values from the literature. We have updated Supplemental Table 2 with the K_m , V_{max} , and K_{eq} values used for modelling. Additionally, we added some further description in the text:

Added text:

“Although we have not measured kinetic parameters for the enzymes used experimentally, our goal was to explore fundamental features of the overall system design. We therefore used kinetic parameters defined for homologous enzymes where needed to obtain a reasonable model. Equations describing enzyme kinetics for glycolysis and the mevalonate pathway were taken from established models describing yeast metabolism³⁵ (SI Data 1) with values for K_m , V_{max} , and K_{eq} based on published values in the BRENDA database³⁶ and previous modelling experiments³⁵. The Gap purge valve (including EcGap, mGap, and NoxE), and PPase were also included in the model. The final parameters and equations are provided in the Supplement.”

4. The authors should also summarize in a separate supplemental table what parameter ranges they checked for the other enzymes that are not mentioned in Fig. 2 in which step size and what their criterion was to decide that the influence was not worth pursuing further.

RESPONSE: We added a more detailed description of the modeling in the Methods section. We feel an additional Supplementary Table would be redundant and possibly confusing.

Added text:

"..a Parameter Scan task was performed for by varying the Vmax of each enzyme or starting concentration of various cofactors (e.g. Pi or ATP) over either a 10 or 50 fold range in 20 steps to identify bottlenecks and optimum starting conditions. Bottleneck enzymes (e.g. PDH) were identified by starting with a 250-500 fold excess over Hex and systematically decreasing each enzyme to a 5 to 10 fold excess followed by monitoring limonene production over 20000 seconds. PDH was the only enzyme that failed to run to completion during the given time frame."

5. The authors might also want to specify further what they mean by "industrially relevant monoterpenes". For example pinene is, to the best of my knowledge, a cheap byproduct of the coniferous wood industry, which is sold for a few dollars per liter (if that much). I doubt that this is more than an illustration product?

RESPONSE: We agree that limonene, pinene, and sabinene are mainly proof-of-concept products. However, there are industrial uses for these compounds that extend beyond flavors and fragrances (ie solvents and biofuel). For use of limonene and pinene in the commodity chemical space/biofuel space, alternative production platforms besides natural sources would be needed to increase supply. Some clarification was added in the introduction.

Added text:

"Even in the case of monoterpenes limonene and pinene where established supply streams exist, their wider adoption as bio-safe solvents and biofuels is limited by amounts produced in nature."

Reviewer #2 (Remarks to the Author):

However, I do have a few quibbles:

1. The authors probably appreciate that terpene synthases typically have very low turnover rates. Moreover, the reactions catalyzed by these enzymes are more akin to little explosion – rip off a diphosphate from the allelic substrate, GPP, with a highly reactive carbocation running around within the active site. There is some evidence in the literature that these enzymes actually catalyze maybe 2 to 4 cycles of catalysis in vitro before pooping out. Looking at the incubation conditions, there isn't anything to suggest the authors have done anything special to "protect" the enzymes. So, what I would like the authors to comment about is how much terpene synthase enzymes they are adding and what percentage of the enzyme is actually staying active during this time frame of days. Perhaps a calculation for the number of turnovers needed for the final yields in relationship units of activity added would be informative. I actually expect that they have to add a stoichiometric excess of the terpene synthases in order to observe the levels of monoterpenes produced. Some additional comments/calculations would be really helpful,

especially because those who have used these enzymes in microfluidic devices tend to see the enzymes die within minutes to hours. What's the trick here?

RESPONSE: We love the reviewer's entertaining description of the reaction. We have now added the total turnover numbers for all the enzymes in Supplemental Table 3. In the case of the terpene synthases, the total turnovers were 2310 to 4310, so while they are slow, they do not appear to be limited due to very low total turnover numbers.

2. When I examine the experimental conditions a bit more closely, I'm also struck that the experiments are being performed on a 200 μl scale, yet the data is being expressed on a liter basis. If the authors have not confirmed that these assays can actually be scaled to these levels, then I strongly recommend caution in representing the data in these terms.

RESPONSE: This is simply standard practice in the field of biofuels/bioproductions. Everything is expressed in g/L so comparisons are easier. If everyone used different units, we'd all have to do conversions on paper to compare parameters with different methods. A review of current literature on *in vitro* reactions shows that even for experiments performed on a small scale (20 μL -1 mL) the titers are reported in g/L. We have not changed this as it is consistent with the literature and could be confusing.

3. Also need more clarification about the overlay – what is the solvent and is there any special means for overlaying the solvent?

RESPONSE: Thank you for pointing out this oversight. We have added a description of the overlay to the Methods section.

Added text:

"...followed by carefully overlaying the reaction sample with 600 μL isopropyl myristate. The reactions were allowed to proceed for at least 24 hours with gentle shaking with limonene production monitored by GC-FID."

4. Probably should provide more background on the various mutants used and give a reference for each

RESPONSE: Thank you again for pointing out this oversight. We have added references to Supplemental Table 1 that describe creation and characterization of mutants used in this study.

5. My most critical comment concerns the efficiency calculations. The authors are attempting to distinguish the Synthetic Biochemistry platform relative to the Synthetic Biology ones. And the calculation is for input glucose conversion to end-product monoterpene. In the Synthetic Biochemistry system, this is a relatively trivial calculation and improving efficiency is a matter of manipulating the input enzymes and co-factors. But what is not being considered in these calculations is the "cost" of the enzymes. These were all produced in bacteria and there isn't any consideration for the carbon feedstock used to grow those bugs. If one were to take those "costs" into consideration, I suspect the efficiency yield calculation would decline by an order of magnitude. The Synthetic Biology platform is appealing because it does not require all the effort of isolating enzymes, qualifying the isolated enzymes, and preserving them for use of days,

months and years. Stocking only the bugs in a freezer serves to preserve the Synthetic Biology system for long-term future use. There are many such reasons why the Synthetic Biology platform has appeal and offer major advantages over the Synthetic Biochemistry platform.

RESPONSE: The reviewer makes a very good point. Synthetic Biochemistry is not currently at the point where it can challenge Synthetic Biology in the production of many bio-based commodity chemicals. We envision a future, however, where all the enzymes we employ will be stable for months or even years. At that point all the concerns about enzyme costs in dollars, energy and glucose will become insignificant. We believe that is very doable and will be a major focus of ours in the near future, now that we have established that these systems can run sustainably.

We have updated the Discussion to reflect the reviewer's sentiments and downplay the ability to immediately implement Synthetic Biochemistry as an alternative to Synthetic Biology:

Added text:

"Implementing a more stable system with engineered enzymes that last for weeks or months will significantly lower the cost of the enzymes not only in terms of dollars but also in initial input of glucose needed to produce enzymes (which was not taken into consideration for our yield calculations). The longer the proteins last (i.e., higher total turnover number), the larger the reduction in up-front costs, including glucose. Our monoterpene platform demonstrates that in vitro systems can reach yields and titers required for industrial production of specialty bio-based chemicals but significant improvements are necessary in areas such as stability and total turnover number before Synthetic Biochemistry can begin to challenge Synthetic Biology for production of commodity chemicals from biomass."

6. Overall, the experimental work presented has been done well and the story line is succinct and easy to digest. However, I do not buy that the Synthetic Biochemistry platform lends itself to large-scale, commodity chemical production. There is a niche for this platform and it may be small scale, perhaps focused on screening mutant enzymes for novel chemistry. Even for a short communication, this reviewer feels it is incumbent upon the authors to fairly represent their work relative to the competing technology.

RESPONSE: We understand the reviewer's skepticism, and we are clear-eyed about the challenges, but we fundamentally disagree. We believe the niche for Synthetic Biochemistry is *exactly* in the production of commodity chemicals and biofuels, where it is critical to obtain high yields, productivities and titers to lower costs. Billions of dollars and huge efforts have already been directed at engineering cells to make biofuels. So where are the next generation biofuels? It turns out it is really hard to balance the needs of cell viability with our needs to reach stringent production parameters. Perhaps Synthetic Biology approaches will ultimately get there, but why not try a different approach?

With an effort that is miniscule relative to Synthetic Biology, we already know that in some cases, Synthetic Biochemistry can beat yields and productivities obtained in cells so far. All we need now is highly stable enzymes to get to still another level. This is exceedingly doable. Although Synthetic Biology seems very appealing in theory and Synthetic Biochemistry might seem too finicky for an industrial process, we believe the reality is exactly the opposite. A set of rock hard enzymes will ultimately be more industrially tough and cheaper to produce than persnickety cells.

As noted in 5 above, we have added the appropriate caveats that Synthetic Biochemistry can't yet compete with Synthetic Biology at scale, but we don't feel it is correct to say that it will not be

useful for industrial chemical production down the road. Give us a chance. We intend to prove the skeptics wrong.

Reviewer #3 (Remarks to the Author):

Concerns:

1. While their work is convincing, a full time course should be reported for figure 5. They mention a key limitation is their rates: “While the flux is still about an order of magnitude below rates needed for industrial production (0.1 g/L/hr vs 1-2 g/L/hr), this work significantly improves upon a previous, less complicated system from PEP to isoprene in both rate (0.014 g/L/hr) and overall extent of reaction (e.g. titer and scale).” However, they don't provide us with data to visualize this ourselves. A full time course of the 7-day reactions for each monoterpene would significantly enhance the reader's ability to observe this key limitation and raise the impact of the work.

RESPONSE: We don't understand this comment as a full 5-day time course for limonene production is already presented in Fig. 4A and B at two different concentrations of glucose, performed in triplicate. We believe these results well describe the course of the reaction.

2. Along these lines, I agree that the flux/rates through their pathway are low. The authors should more comprehensively compare these rates to other in vitro systems, both synthetic enzymatic pathways as well as lysates? What are strategies to improve the rates? Also, what do rates look like for different classes of molecules in various purified and lysate systems?

RESPONSE: Thank you for this suggestion. Natural terpene synthases are slow enzymes so this is a major limitation for both in vivo and in vitro production of terpenes. As there may be little evolutionary pressure on terpene synthases to be fast, it seems possible that rates could be improved by directed evolution or engineering. We added some discussion of this and comparison to other in vitro systems in the Conclusion section.

Added text:

While the flux is still about an order of magnitude below rates needed for industrial production (0.1 g/L/hr vs 1-2 g/L/hr), this work significantly improves upon a previous, less complicated system³⁸ from PEP to isoprene in both rate (0.014 g/L/hr) and overall extent of reaction (e.g. titer and scale). Rates for in vitro production of less complex molecules such as ethanol⁴⁴, 2,3 butanediol⁴⁵, mevalonate³⁰, and polyhydroxybutyrate³² from glucose in either a lysate or purified system range from 0.7 g/L/hr to as high as 47.9 g/L/hr, suggesting that the rate for in vitro terpene production could be raised if either more terpene synthase were employed or the rate was significantly improved through engineering. Therefore, with further optimization of reaction conditions as well as in vitro evolution of enzyme stability, and activity (e.g. terpene synthase) it should be possible to develop a faster system that functions for much longer.

3. Additionally, please represent figure 5 with appropriate distribution of alpha versus beta pinene. Also describe the relative amounts sabinene compounds (instead of summing total monoterpenes). The current plot implies that pure compounds are produced. Please comment how this system affects the ratio of alpha and beta pinene compared to reference 39.

RESPONSE: Thank you for this comment. We previously showed the distribution in the supplement, but we now provide the full quantified distribution to the Fig. 5 legend and point out that the pinene and sabinene synthases produce a mixture in the main text.

Revised main text under “Production of Other Monoterpenes from Glucose”:

“To test whether our synthetic biochemistry system could be used as a broad platform for the production of other monoterpenes, we attempted to produce pinene, and sabinene from glucose. To make pinene or sabinene, the appropriate monoterpene synthase was simply swapped with the limonene synthase used above. For pinene synthesis, α -pinene synthase from *Picea sitchensis* was used which has been shown to produce a mixture of 77% α -pinene and 22% β -pinene⁴⁴. For sabinene production the limonene synthase mutant N345A was used which produces sabinene as its major product along with smaller amounts of other monoterpenes⁴⁵. Starting with 500 mM glucose, monoterpene production was allowed to proceed for 7 days and then quantified by GC (Supplementary Figure 3). Pinene titer (sum of α -pinene and β -pinene nearly identical to distribution in ref. 44) reached 14.9 ± 0.6 g/L while sabinene titer (total monoterpenes, similar distribution to ref. 45) was 15.9 ± 0.4 g/L with yields of $103.9\% \pm 8.1\%$ and $94.5\% \pm 4.7\%$ of theoretical respectively (Figure 5). Like limonene production, these values are at least an order of magnitude higher than microbial production for pinene¹⁴ and sabinene¹⁵ and multiple times higher than the toxicity limit for these compounds^{15,43}.”

Added text to Figure 5 legend:

“For the limonene synthase reaction (green bar), the product was 100% limonene. For the pinene synthase reaction (purple bar), the product distribution is $80.8 \pm 0.33\%$ α -pinene and $19.2 \pm 0.33\%$ β -pinene. For the sabinene synthase reaction catalyzed by the N345A Limonene synthase mutant (orange bar), the product distribution is $11.5 \pm 0.03\%$ α -pinene, $11.2 \pm 0.01\%$ β -pinene, $37.9 \pm 0.02\%$ (-)-sabinene, $3.5 \pm 0.01\%$ (+)-sabinene, $27.1 \pm 0.04\%$ limonene, and $5.7 \pm 0.01\%$ myrcene.”

4. In Figure 3, the authors show a very simple +/- system of the valve enzymes. I am curious to see whether there are optimal amounts of these enzymes together. Are there concentrations of NoxE for example that still yield limonene but are low? Are there concentrations that are too high? How robust is this valve? While the authors use their computer model to identify bottlenecks and only choose to focus on the most sensitive parameters to emphasize robustness, not optimums, it would nice to see a sensitivity analysis or optimization of their new purge valve (as this is a key claim of their work)

RESPONSE: We appreciate the reviewer’s interest in the purge valve. We are partial to the concept as well. However, we extensively characterized the robustness of the purge valve system in our original work (Ogpenorth, et. al. Nat Comm **5**, 4113 (2014)), which was subsequently reinforced by the demonstration and use of two new purge valves in a follow up paper in Nature Chemical Biology (Ogpenorth, et. al. **12**, 393-395 (2016)). At this point we feel that the purge valve concept no longer has novelty and is simply a tool that we use in synthetic biochemistry systems.

5. Additionally, how was the level of glutathione reductase chosen? This reaction consumes NADPH but the authors do not comment on its role in the system and how it interacts with the purge valve system.

RESPONSE: This is an interesting point and exactly why our purge valve system is important. Our overall system design generates an excess of NADPH, while still allowing carbon flux via the purge valve, which allows for enough NADPH production to tolerate spontaneous oxidation of NADPH and other minor side reactions like this one. In the current implementation, glutathione reductase and catalase were simply added as a precaution prevent oxidation over the course of 7 days. We did not optimize or investigate the need for these components. The need for these activities will ultimately depend on the enzymes used and we certainly don’t envision that the enzymes used in the current system will necessarily find their way to an industrial process.

We added the following statement in the Methods section. We thank the reviewer for helping us to highlight another advantage of our purge valve concept.

Added text:

"Glutathione reductase and catalase were simply added as a precaution prevent oxidation over the course of 7 days. We did not optimize or investigate the need for these components. As the overall system design with the purge valve produces an excess of NADPH, there is sufficient NADPH in the system to supply glutathione reductase."

6. The system is described as "robust" in line 120 and 171. I suggest a brief sensitivity analysis to enzyme concentration and cofactor concentration [possibly also reaction temperature, volume, etc] to support this statement.

RESPONSE: We agree that using the term "robust" may be too strong to describe the current system (although it does function for at least 5 days). As a result, we removed the word "robust" and replaced it with the word "efficient".

7. Line 94...The authors state that "cell free systems have avoided the use of ATP for building chemicals" and cite a 2012 review by Guterl et al. However, high ATP turnover has been observed in a number of cell free systems, both purified and crude extracts, for synthesis of metabolites and proteins (e.g., Ye, X., Honda, K., Sakai, T., Okano, K., et al., Synthetic metabolic engineering-a novel, simple technology for designing a chimeric metabolic pathway. *Microb. Cell Factories* 2012, 11, 120 and others). Can the above statement be expanded to convey the idea that ATP regeneration has been used in cell-free systems before?

RESPONSE: Thank you for this comment. We have now included a more complete description of how ATP has been utilized in the past for in vitro systems.

Added text:

" So far complex cell free systems for building chemicals have either avoided the use of ATP completely³⁰ or stoichiometrically balance ATP use and consumption during catabolic reactions only³⁵. While ATP specific regeneration systems have been developed, these are generally simple systems consisting of only a few enzymes that utilize sacrificial substrates (e.g. polyphosphate or acetyl-phosphate)^{36,37}. It remains unclear whether excess ATP (e.g. the net 2 ATP produced from glycolysis) can be effectively employed outside the cell to drive anabolic pathways in a fully recyclable manner^{30,31}"

8. While beyond the scope of this work, I am curious to know what are the limits of the current system? Why does the reaction stop? Is fed-batch feeding of glucose possible?

RESPONSE: We believe the reaction stops due to instability of one or more of the enzymes used. We are currently working on a follow-up study to employ more stable enzymes that we hope will enable the system to function for much longer. There is no reason why fed-batch is not possible however logistically, a single, high concentration addition is easier. For the 7 day reaction there is still glucose remaining suggesting the reaction stops for reasons other than running out of substrate as is shown in the time course in figure 4.

9. Please add a brief comment at line 98 or 159 to clarify that this purge valve is at the Gap step in contrast to previous work that made a purge valve at the PDH step (reference 31). This will help readers not intimately familiar with prior literature in the field.

RESPONSE: We have added clarification to differentiate this from the prior work:

Added text:

"...(rather than the PDH step described previously³²)"

Reviewer #1 (Remarks to the Author)

I feel that the authors have adequately addressed all my concerns. Therefore, I would like to reiterate that I think that this is a major advance over prior art and perfectly suitable for publication in Nature Communications.

Reviewer #2 (Remarks to the Author)

The authors have adequately addressed my previous concerns, and appear to have also done so for the other reviewer comments. There is no question that this work will stimulate lots of consideration of the Synthetic Biochemistry approach. I remain skeptical about its potential for commodity chemical scale production, but am open to be convinced otherwise.

Reviewer #3 (Remarks to the Author)

The authors have done a nice job responding to reviews by all referees. The improved manuscript makes their achievements even that much more impressive. I support publication but have one minor remaining issue in Figure 5 that should be addressed.

While I appreciate the author's revision of the figure legend in Figure 5, I believe the plot axis should also be changed. It is misleading to label the x-axis "sabinene" and draw a bar at ~16 g/L when the actual (-)-sabinene titer is I believe $(37.9)/100 * 15.9 = 6.02$ g/L. The logic is similar for the pinene bar. The author's current representation of their data implies that the N345A mutation to limonene synthase produces a single sabinene product which is not the case. At minimum, the y-axis should be labeled "net monoterpene (g/L)" and the x-axis labels changed to say "limonene synthase", "pinene synthase" "limonene synthase N345A."

Response to Reviewer Comments

Reviewer #3 (Remarks to the Author):

Concerns:

1. While I appreciate the author's revision of the figure legend in Figure 5, I believe the plot axis should also be changed. It is misleading to label the x-axis "sabinene" and draw a bar at ~16 g/L when the actual (-)-sabinene titer is I believe $(37.9)/100 \times 15.9 = 6.02$ g/L. The logic is similar for the pinene bar. The author's current representation of their data implies that the N345A mutation to limonene synthase produces a single sabinene product which is not the case. At minimum, the y-axis should be labeled "net monoterpene (g/L)" and the x-axis labels changed to say "limonene synthase", "pinene synthase" "limonene synthase N345A."

RESPONSE: We have relabeled FIGURE 5 so the y-axis now reads "net monoterpenes (g L-1)" and added the identity of the synthase used in parentheses on the x-axis below each bar.